# Combined Use of Three-Dimensional Construction and Indocyanine Green-Fluorescent Imaging for Resection of Multiple Lung Metastases in Hepatoblastoma

**DOI:** 10.3390/children9030376

**Published:** 2022-03-08

**Authors:** Shugo Komatsu, Keita Terui, Mitsuyuki Nakata, Ryohei Shibata, Satoru Oita, Yunosuke Kawaguchi, Hiroko Yoshizawa, Tomoya Hirokawa, Erika Nakatani, Tomoro Hishiki

**Affiliations:** Department of Pediatric Surgery, Chiba University Graduate School of Medicine, 1-8-1 Inohana, Chuo-ku, Chiba 260-8677, Japan; kta@cc.rim.or.jp (K.T.); mitchinakachi@gmail.com (M.N.); shibataryohei@gmail.com (R.S.); z5m1019@gmail.com (S.O.); yunosuke.kawaguchi@gmail.com (Y.K.); junjuni_20koropanda@yahoo.co.jp (H.Y.); ayda4475@chiba-u.jp (T.H.); magnolia.eri29@gmail.com (E.N.); hishiki@chiba-u.jp (T.H.)

**Keywords:** hepatoblastoma, multiple lung metastases, three-dimensional imaging, indocyanine green

## Abstract

It is essential to accurately and safely resect all tumors during surgery for multiple lung metastases. Here, we report a case of hepatoblastoma (HB) with multiple pulmonary nodules that ultimately underwent complete resection using combined three-dimensional image reconstruction and indocyanine green (ICG) fluorescence guidance. A 1-year-old boy was diagnosed with HB and multiple lung metastases. After intensive chemotherapy, complete resection with subsegmentectomy (S5 + 6) and partial resection (S3, S8) were performed. More than 100 pulmonary nodules, which remained visible on computed tomography (CT) despite additional postoperative chemotherapy, were subjected to pulmonary resection. We used the SYNAPSE VINCENT software (Fujifilm Medical, Tokyo, Japan) to obtain three-dimensional images of the nodules. We numbered each nodule, and 33 lesions of the right lung were resected by multiple wedge resections through a right thoracotomy, with the aid of palpation and ICG fluorescence guidance. One month after the right metastasectomy, resection of 64 lesions in the left lung was performed via left thoracotomy. Postoperative CT showed complete clearance of the lung lesions, and the patient remained disease-free for 15 months after the treatment. This case study confirms that the combination of three-dimensional localization and ICG fluorescence guidance allows for accurate and safe resection of nearly 100 lung metastases.

## 1. Introduction

Hepatoblastoma (HB) is the most common malignant liver tumor in children. Approximately 10–20% of patients with HB have distant metastases at the time of initial diagnosis, mostly in the lungs. The outcomes of HB have been improved dramatically with the advancement of multidisciplinary treatments using cisplatin-based chemotherapy and surgery [1]. However, the presence of distant metastasis at diagnosis remains the most powerful predictor of poor prognosis. The 5-year event-free survival rate is less than 60% in patients with pulmonary metastasis [2]. Approximately 30–50% of patients with lung metastases at diagnosis achieve clearance of nodules with chemotherapy alone [3,4,5]. For chemotherapy-refractory lung metastases, complete resection can be expected to maintain remission if the primary tumor is completely resected and there are no other metastatic sites [3,6,7,8]. Although there are no definite indications for resection of pulmonary metastases, they are generally restricted to nodules that remain after repeated courses of chemotherapy. Moreover, no studies to date have directly compared surgical and non-surgical approaches for residual lung metastases in a prospective setting, and some reports have suggested that removal of residual metastases may improve the prognosis and resection of metastases should be considered more aggressively [3,6,7]. However, the safety and accuracy of surgery can be problematic in patients with multiple lung metastases. There is also a risk of postoperative respiratory dysfunction due to decreased residual lung capacity and risk of recurrence due to tumor remnants.

Wedge resection, which is used to resect tumors while preserving lung function, is generally approved as the standard approach for peripheral nodules, especially for metastatic tumors. Accurately establishing the site of the nodules, preoperative identification of anatomical changes, and preoperative surgical process planning are hurdles in wedge resection. A more accurate preoperative evaluation is required when considering surgical indications that are close to the safety limit. Conventional computed tomography (CT) imaging is the gold standard for the preoperative planning of pulmonary metastasectomy. Three-dimensional CT lung reconstruction can accurately determine the relative positions of nodules in the lobes; therefore, promising techniques using three-dimensional simulation have been developed in recent years [9,10].

Early three-dimensional CT imaging software was unable to accurately reconstruct pulmonary veins, arteries, and bronchi, as well as automatically partition pulmonary segments and identify the exact location of lung nodules [11,12]. However, these problems have been resolved by upgrading the reconstruction level. The SYNAPSE VINCENT imaging reconstruction software can build three-dimensional images of lung segments and nodules using high-resolution CT. The partition of pulmonary segments, location of the nodules, and space between nearby structures, pulmonary veins, and bronchi may all be visualized using this software for surgical planning. The application of three-dimensional simulation technology has made new anatomical and morphological approaches possible and contributed to the accuracy of surgery. However, despite advances in imaging technology, it is difficult to exclude the possibility of malignancy based solely on CT findings. Therefore, in surgery for pulmonary HB metastases, exploration through thoracotomy is generally recommended for full palpation of unrecognized nodules.

Preoperative image evaluation and intraoperative visual and palpatory examinations have been traditionally performed for pulmonary HB metastases; however, the real-time identification of cancer tissues is critical. The outstanding visibility of HB under near-infrared light (750–810 nm) has been observed in several pilot investigations [13,14]. Indocyanine green (ICG) emits light with a peak at approximately 840 nm when irradiated with near-infrared light (750–810 nm). ICG-guided surgery has been widely applied to detect both liver and metastatic site lesions in HB because of its unique properties, especially its ability to accumulate in the liver [15,16,17]. When ICG is administered intravenously, it is taken up by hepatocytes and excreted in the bile. Since HB is a hepatocyte-derived tumor, ICG is taken up by tumor cells and remains there for a long time, owing to its disrupted excretion mechanism of action. Therefore, the difference in ICG excretion times between normal and tumor cells can be used to visualize the tumor area [18]. There are two types of commercial detectors for near-infrared fluorescence devices: handles and endoscopes. The handle type fluoresces more intensely than the endoscopic type, even in the same lesion [19]. The choice of device is important; the handle type is generally used in open abdominal and thoracic surgeries, whereas the endoscopic type is often used in endoscopic surgery [20]. The main advantages of ICG fluorescence imaging in a clinical setting are its safety and practicality. The incidence of adverse reactions has been reported to be less than 0.01% [21]. ICG fluorescence can be used to confirm the presence or absence of residual tumors in the liver at the margin of the hepatic resection in primary lesions [16]. In metastases, even very small foci can be visualized because of the high contrast. Furthermore, ICG fluorescence is useful in multiple lung-metastasis surgery for confirming the location of lesions and preventing missed lesions, which previously had to be performed by visual and palpation examinations [20].

Herein, we present a surgical case of HB with >100 resistant pulmonary metastases. We attempted to perform the surgery using a combination of three-dimensional image simulation technology and ICG fluorescence navigation, rendering the surgery accurate and safe.

## 2. Case Report

A 1-year-old boy was referred to our hospital with a chief complaint of an abdominal mass. A blood test showed an elevated alpha-fetoprotein (AFP) level of 163,622 ng/mL, and CT confirmed the presence of a large hepatic mass involving segments 3, 5, 6, and 8 of the liver and multiple nodular shadows in the bilateral lungs (Figure 1).

The patient was diagnosed with PRETEXT III HB (V−, P−, E−, F−, R−, C−, N−, M+) with multiple lung metastases. Dose-dependent, cisplatin-based intensive chemotherapy was administered according to the SIOPEL-4 treatment protocol [4]. Preoperative magnetic resonance imaging showed that the primary tumor size was reduced to 7 cm. In addition, 4 mm and 6 mm-large masses were detected in S3 and S8 of the liver, respectively. A blood test showed a decreased AFP level of 301 ng/mL. Based on these findings, complete resection through subsegmentectomy and enucleation were performed for surgical removal of S5 and S6, and S3 and S8, respectively. The pathological findings of the primary lesion suggested that half of the original tumor tissue had disappeared. In addition, tumor cells were found in S3 where enucleation was performed, but no tumor cells were detected in the resection specimen of S8. Postoperative chemotherapy did not achieve complete remission of the pulmonary nodules. There were 45 and 73 nodules in the right and left lungs, respectively, that remained visible on CT and were subjected to pulmonary resection. We used the SYNAPSE VINCENT software (Fujifilm Medical, Tokyo, Japan) to obtain three-dimensional images of the nodules. We then numbered all nodules detectable on the CT image. We manually marked and numbered each nodal region in the two-dimensional CT image using the software and subsequently constructed a three-dimensional image (Figure 2).

The metastatic lesions were resected through right and left thoracotomies and performed separately, with an interval of 1 month, since the thoracotomies were planned in between the chemotherapy treatments, according to the protocol. The level of AFP decreased to 23 ng/mL and to 15 ng/mL, respectively, before the resection of the right and left lung metastases. The affected lung was collapsed with isolated ventilation. ICG (Diagnogreen, Daiichi-Sankyo Pharma, Tokyo, Japan) was administered at a dose of 0.5 mg/kg 24 h before surgery. Since ICG is often used off-label, informed consent was obtained from the patient’s guardian after approval from the hospital’s ethics committee. Lesions corresponding to the numbered nodules on the three-dimensional-constructed images were thoroughly searched by careful palpation, and an ICG fluorescence imaging system (Photodynamic Eye, Hamamatsu Photonics, Hamamatsu, Japan) was used as a reference to identify the lesions (Figure 3).

We searched for all lesions using palpation and ICG fluorescence, based on the number assigned to each nodule in the three-dimensional images we constructed. All nodules palpated intraoperatively were removed, despite not being ICG-positive. None of the CT-detected nodules that could not be palpated intraoperatively showed ICG fluorescence. Thirty-three lesions detected in the right lung were resected via right thoracotomy. All nodules were located within 6 mm from the pleural surface and thus could be removed through wedge resections. Nodules were resected using margins of approximately 1–5 mm, and the defects were repaired with interrupted Z-shaped 4-0 PDS sutures. We avoided the use of staplers since this would result in excess removal of the surrounding normal lung tissue. One month after the right metastasectomy, resection of 64 lesions in the left lung was performed via left thoracotomy, with one course of chemotherapy in between. In both lung operations, 73 wedge resections were performed, and 97 lesions were removed. Both surgeries were completed without complications, and the postoperative recovery was quick. Histological examination demonstrated that 53 of the 97 removed lesions contained tumors, but the degree of differentiation was higher than that of the primary lesion in the liver. In addition, there were several specimens that showed fibrosis and foam cells, which were presumed to be traces of tumor cell disappearance, suggesting that a certain therapeutic effect of chemotherapy had been achieved. Of the resected lesions, 74 were fluorescent-positive lesions, and 52 were HBs. Overall, 23 lesions were fluorescent-negative, and 22 did not contain metastases on histopathology (Table 1).

To ascertain the intraoperative fluorescence of ICG, we further investigated the ICG fluorescence of all resected specimens using a fluorescence microscope (BZ-X800, Keyence, Osaka, Japan) with a near-infrared filter set (49,030, Chroma Technology, Bellows Falls, VT, USA). Observation of lesions pathologically diagnosed as lung HB metastases showed ICG fluorescence in almost all tumor cells. None of the 22 lesions that were false-positive for ICG intraoperatively showed ICG fluorescence using fluorescence microscopy. However, there were several specimens in which fibrosis, foam cells, and hemosiderin-phagocytosing macrophages were observed with hematoxylin and eosin staining. In contrast, only one lesion that was false-negative on intraoperative ICG fluorescence was fluorescent in the pathological tissue (Figure 4).

Postoperatively, AFP remained at normal levels, and a CT scan obtained 1 year after surgery showed complete clearance of the lung lesions. Because it was considered difficult to perform a detailed evaluation in the early postoperative period due to surgical artifacts, CT was performed for the first time one year after the surgery. The patient remained disease-free 15 months after the last treatment. The patient was faring well without respiratory symptoms.

## 3. Discussion

ICG imaging is effective for the resection of pulmonary HB metastases. However, few studies have examined ICG fluorescence using resected specimens [16,20]. We used fluorescence microscopy to determine the intraoperative ICG fluorescence of the resected specimens. Microscopic examination of the tumor cells confirmed ICG fluorescence in almost all tumor cells. Conversely, no fluorescent structures other than tumor cells were observed. We investigated the number of tumor cells that were accumulated and could be visualized intraoperatively. In our case, we also found ICG fluorescence intraoperatively in a tumor 1.1 mm in size, which was the smallest of all resected lesions. Furthermore, the smallest previously reported nodule was 0.062 mm in size [13]. Thus, ICG fluorescence imaging is a useful method for detecting tiny metastatic HB tumors that are neither palpable nor visible.

Another advantage is that ICG is characterized by very high sensitivity [13]. In our case, ICG fluorescence was also observed in 52 of the 53 pathologically diagnosed tumor lesions, with a sensitivity and specificity of 98% and 50%, respectively. Although ICG-guided surgery is a highly sensitive method for detecting pulmonary HB metastases, it is associated with a high proportion of false-positive results. Therefore, it is critical for surgeons to avoid unnecessary resection based on false-positive fluorescence results. Yamada et al. reported that there are two reasons for false-positive outcomes in vivo: the first one results from surgeons mistakenly assigning non-fluorescent spots as fluorescent spots, and the second results from the fact that some non-cancerous lesions do indeed take up ICG to some degree [16]. Large regenerative nodules, bile duct proliferation, dysplastic nodules, chronic inflammation, fibrosis, normal liver parenchyma, bile plugs, and cysts have been reported as ICG-positive non-malignant pathological observations in the liver [16]. In the lungs, inflammatory cells, hemorrhage, thrombosis, necrotic tissue, granulomas, alveolar cells, and Vicryl sutures have been reported as ICG-positive non-malignant pathological observations [13,20]. In our case, the false-positive rate was 30%, and none of the false-positive lesion specimens showed ICG fluorescence using fluorescence microscopy. However, hematoxylin and eosin staining showed that some of the false-positive specimens included fibrosis, foam cells, and hemosiderin-phagocytosing macrophages. These findings suggest that tumor cells may have once been present there. In addition, it is possible that the false-positive results involved small tumors that did not fit well into the specimen section. To clarify this, we would need to make whole sections of each specimen, which was practically impossible in this study because of the large number of lesions.

Another technical pitfall is that the near-infrared light penetration depth of human tissue is limited to 5–10 mm [22]. Consequently, current devices miss deeper lesions in the tissue. In our case, fluorescence microscopy of the only false-negative lesion showed ICG fluorescence. The tumor was located approximately 5 mm from the pleural surface, suggesting that the depth of the lesion may have influenced the false-negative outcome. In contrast, the low number of tumor cells contained in the specimen suggests that the threshold for visualization of intraoperative fluorescence may have not been reached, but the number of tumor cells in false-negative lesions observed was not lower than that of other tumor lesions that were ICG fluorescent.

Extraction of a very large number of lung metastases was required in our case; therefore, three-dimensional constructed images were very useful. Each nodule was numbered and inspected intraoperatively, which allowed us to search for all lesions detected by CT without missing any. Though there are reports on resection methods for micro-lung metastasis lesions, such as CT-guided marking [23,24], it is difficult to adapt for multiple lung metastasis lesions, such as in our case. Surgery is rarely indicated for hepatocellular carcinoma and other tumors when numerous lung metastases are observed. There are few reports on valuable methods for multiple lung metastases. However, in HB, total resection of lung metastases should be considered as seriously as possible since it is expected to improve the prognosis. To our knowledge, there have been no reports of surgery using three-dimensional constructed images for multiple lung metastases of HB. Our approach of using three-dimensional construction images is novel, useful, and recommended for the future.

Although there was a slight discrepancy between the number of nodules observed on the preoperative CT and the number of lesions resected on surgery, the tumor had disappeared entirely on the CT one year after surgery. The possible reasons for this were that the lesions were non-neoplastic and exhibited inflammatory changes; they disappeared spontaneously because of chemotherapy between the CT imaging and surgery; or they did not fit well in the specimen section but were included in the removed specimens.

In recent years, several attractive fluorescent probes that can label cancerous tissues in vivo have been developed; however, these compounds cannot currently be administered to patients in a clinical setting [25,26,27,28]. ICG has a well-established safety profile and hepatobiliary drainage, making ICG fluorescence a promising method for imaging hepatobiliary diseases. However, new optical techniques that allow for deeper studies and automated standardization are desired.

## 4. Conclusions

Surgery should ideally be both curative and safe. This case study confirmed that the combination of three-dimensional localization and ICG fluorescence guidance was helpful in accurately localizing each nodule detected on preoperative imaging and safely resecting nearly 100 lung metastases. These techniques should be considered the first option for safer and more reliable resections of multiple lung metastases.

## Figures and Tables

**Figure 1 children-09-00376-f001:**
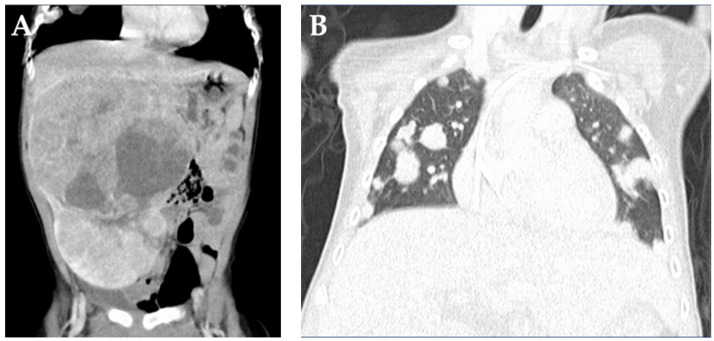
Computed tomography findings on admission. (**A**) A large tumor is located in the right lobe. (**B**) Multiple metastatic pulmonary nodules are observed in both lungs.

**Figure 2 children-09-00376-f002:**
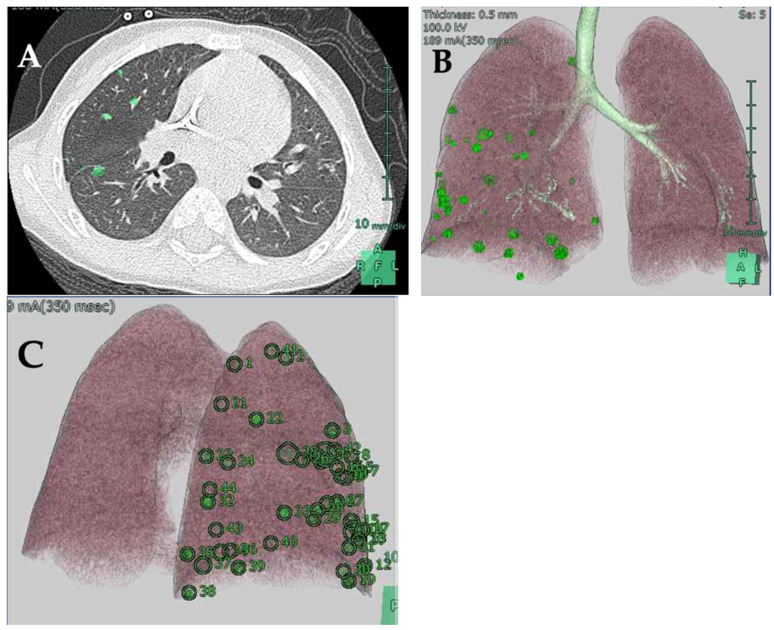
Three-dimensional image of the pulmonary nodules using the SYNAPSE VINCENT software. (**A**) Each nodal region is manually marked on a two-dimensional CT image. (**B**) A three-dimensional image of pulmonary nodules. (**C**) Three-dimensional image with each metastatic site numbered.

**Figure 3 children-09-00376-f003:**
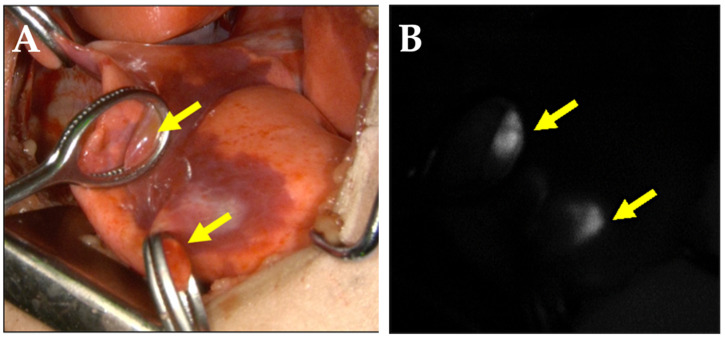
Metastatic lung lesions. (**A**) Visible ray photograph demonstrating the intraoperative lung surface. Yellow arrows indicate lung metastases. (**B**) Photograph demonstrating the lung metastases of the lesions shown in (**A**) using fluorescence.

**Figure 4 children-09-00376-f004:**
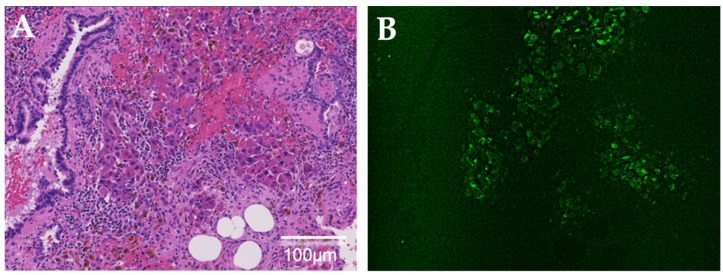
Histological examination via false-negative indocyanine green (ICG) fluorescence on intraoperative observation. (**A**) Lung metastatic cells of hepatoblastoma are identified in the hematoxylin and eosin-stained specimen. (**B**) ICG fluorescence is observed using a fluorescence microscope.

**Table 1 children-09-00376-t001:** Sensitivity and specificity of indocyanine green (ICG) fluorescent imaging test in the pulmonary metastases resections of hepatoblastoma.

ICG Fluorescent Imaging	Histological Examination	
	Tumor	Non-Tumor
Positive	52	22
Negative	1	22

## Data Availability

The data presented in this study are available upon request from the corresponding author. The data are not publicly available because of patient privacy.

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
