# Peer review of "Combined Use of Three-Dimensional Construction and Indocyanine Green-Fluorescent Imaging for Resection of Multiple Lung Metastases in Hepatoblastoma"

_children, 2022, doi:10.3390/children9030376_

Round 1
Reviewer 1 Report
The article is devoted to an actual question - the hepatoblastoma with lung metastases management. Today in cases when metastases are not cleared after chemotherapy, metastasectomy is the last curative option. But sometimes complete removal of these lesions is a difficult challenge due to their intraparenchymal location, shrinkage after chemotherapy, and softness of the lesions. Last few years indocyanine green fluorescent (ICG) imaging was developed as a perspective method for intraoperative navigation for liver resections and metastases removal for hepatocellular carcinoma and hepatoblastoma. Despite the existence of several case reports with a few patients which underwent metastasectomy with ICG this article has an undoubted advantage due to confirming the possibility of complete removal of metastatic lesions even with their significant number (97). So this case report describes the usefulness of fluoroscopy with ICG and encourages clinicians to solve the problem of hepatoblastoma metastases treatment.
Despite of the undeniable advantages of this case report, I believe correction of some details can improve the article and increase its scientific significance for future research:
- In this case the PRETEXT stage was 3, but according to preoperative computed tomography (CT) the segments 3,5,6 and 8th were involved which corresponds to the 4th PRETEXT stage. Because PRETEXT or POSTTEXT 4 can be a contraindication to one-stage liver resection the preoperative examination and surgery details should be reported.
- There remained 118 foci after chemotherapy according to CT (73 - in the right lung and 45 in the left). Nevertheless only 97 lesions were removed. It should explain the possible reasons for this discrepancy and the results of computed tomography (CT) after surgery. The CT result in a year after surgery is only described in the article.
- The interval between thoracotomies was 1 month which is too long keeping in mind the diagnosis. On the other hand, the knowledge about postoperative complications frequency and their severity may influent to role of surgery in lung metastases treatment. So I consider that it is important to describe the recovery time, presence or absence of complications and the alpha-fetoprotein level changes during stages of treatment.
- The reported sensitivity high rate (98%) is the advantage of using ICG, depending on the histological type of tumor and the amount of necrotic cells. So the histology details after liver resection and metastasectomy should be written.
- There are no references for statements on page 2 lines 84-85, 85-87, 89-91, 91-94, and on page 5 lines 167-168.
Author Response
Answer to reviewers
We greatly appreciate your suggestions.
The followings are answers to the suggestions.
- In this case the PRETEXT stage was 3, but according to preoperative computed tomography (CT) the segments 3,5,6 and 8th were involved which corresponds to the 4th PRETEXT stage. Because PRETEXT or POSTTEXT 4 can be a contraindication to one-stage liver resection the preoperative examination and surgery details should be reported.
I would like to thank you for your advice. Preoperative magnetic resonance imaging showed that the primary tumor size (S5, S6) was reduced to 7 cm. In addition, a 4-mm- and 6-mm-large masses were detected in S3 and S8 of the liver, respectively. Since the left medial section was free, we decided on POSTTEXT 3. Based on these findings, complete resection through subsegmentectomy and enucleation were performed for surgical removal of S5 and S6, and S3 and S8, respectively.
I would agree with the reviewer’s suggestion and added some explanation as to the following: “ Dose-dependent cisplatin-based intensive chemotherapy was administered according to the SIOPEL-4 treatment protocol [22]. Preoperative magnetic resonance imaging showed that the primary tumor size was reduced to 7 cm. In addition, a 4-mm- and 6-mm-large masses were detected in S3 and S8 of the liver, respectively. A blood test showed a decreased AFP level to 301 ng/ml. Based on these findings, complete resection through subsegmentectomy and enucleation were performed for surgical removal of S5 and S6, and S3 and S8, respectively. (lines 125-131) “
- There remained 118 foci after chemotherapy according to CT (73 - in the right lung and 45 in the left). Nevertheless only 97 lesions were removed. It should explain the possible reasons for this discrepancy and the results of computed tomography (CT) after surgery. The CT result in a year after surgery is only described in the article.
Thank you for your important suggestion. We have three possibilities. The first is that the lesions were non-neoplastic, such as inflammatory changes, and spontaneously disappeared. The second is that the lesions had disappeared due to chemotherapy between CT imaging and surgery. The third is that the lesions did not fit well in the specimen section but were included in the removed specimens.
Because it was considered difficult to perform a detailed evaluation in the early postoperative period due to surgical artifacts, CT was performed for the first time one year after the surgery. There has been no increase in AFP levels postoperatively and we do not expect any recurrence.
I agree with the reviewer's suggestion and added some explanation like the following (lines 349-354): “ Although there was a slight discrepancy between the number of nodules observed on the preoperative CT and the number of lesions resected on surgery, the tumor had disappeared entirely on the CT one year after surgery. The possible reasons for this were that the lesions were non-neoplastic, exhibiting inflammatory changes; they disappeared spontaneously because of chemotherapy between the CT imaging and surgery; or they did not fit well in the specimen section but were included in the removed specimens. “
- The interval between thoracotomies was one month which is too long keeping in mind the diagnosis. On the other hand, the knowledge about postoperative complications frequency and their severity may influent to role of surgery in lung metastases treatment. So I consider that it is important to describe the recovery time, presence or absence of complications and the alpha-fetoprotein level changes during stages of treatment.
I would like to thank you for your advice. All surgeries were completed without complications and the postoperative recovery was quick. The level of AFP was decreased to 301 ng/ml before resection of the primary tumor, 23 ng/ml before resection of the right lung metastasis, 15 ng/ml before resection of the left lung metastasis, and decreased to 7 ng/ml one month after the last surgery. The level of AFP has not been elevated again since then.
As you pointed out, we think that one month is a long interval. Normally, we have a policy of simultaneous bilateral treatment. However, in this case, we thought we should consider left metastasectomy after checking the details of the pathological tissue, e.g., the percentage of viable cells in the removed specimen and the postoperative changes in the level of AFP. Because resection of multiple lung metastases is highly invasive., we had to think carefully about whether it would be useful to perform a left metastasectomy. Based on these considerations, in this case, we decided from the beginning that the metastasectomy would be performed in two stages. Since the surgery was planned in between the chemotherapy treatments according to the protocol, there was one month interval between the surgeries.
We have revised the relevant sections and added recovery time, presence or absence of complications, and changes in alpha-fetoprotein level during treatment stages (lines 118-120, 129-131, 156-160, 195-202).
- The reported sensitivity high rate (98%) is the advantage of using ICG, depending on the histological type of tumor and the amount of necrotic cells. So the histology details after liver resection and metastasectomy should be written.
I would like to thank your advice and explain the following. “ The pathological findings of the primary lesion suggested that half of the original tumor tissue had disappeared. In addition, tumor cells were found in S3 where enucleation was performed, but no tumor cells were detected in the resection specimen of S8. (lines 131-134) “
“ Histological examination demonstrated that 53 of the 97 removed lesions contained tumor, but the degree of differentiation was higher than that of the primary lesion in the liver. In addition, there were several specimens that showed fibrosis and foam cells, which were presumed to be traces of tumor cell disappearance, suggesting that a certain therapeutic effect of chemotherapy had been achieved. (lines 196-200) ”
- There are no references for statements on page 2 lines 84-85, 85-87, 89-91, 91-94, and on page 5 lines 167-168.
Thank you for your suggestion. We added references for each.
lines 84-85 (lines 85−86: revised version):
ï¼»19ï¼½ Kitagawa, N.; Shinkai, M.; Mochizuki, K.; Usui, H.; Yagi, Y.; Yukihiro, T.; Ikkei, O.; Akio, K.; Mio, T.; Tanaka, Y. Examination of Performance of Detection Device in ICG Fluorescence Method: Comparison of Handheld Type and Endoscopic Type. The Japanese Journal of Pediatric Hematology/Oncology. 2021, 58, 292.
lines 85-87 (lines 86-88: revised version):
ï¼»20ï¼½ Hiyama, E. Fluorescence Image-Guided Navigation Surgery Using Indocyanine Green for Hepatoblastoma. Children (Basel) 2021, 8, 1015.
lines 89-91 (lines 90-92: revised version):
ï¼»16ï¼½ Yamada, Y.; Ohno, M.; Fujino, A.; Kanamori, Y.; Irie, R.; Yoshioka, T.; Miyazaki, O.; Uchida, H.; Fukuda, A.; Sakamoto, S.; Kasahara, M.; Matsumoto, K.; Fuchimoto, Y.; Hoshino, K.; Kuroda, T.; Hishiki, T. Fluorescence-Guided Surgery for Hepatoblastoma with Indocyanine Green. Cancers (Basel) 2019, 11, 1215.
lines 91-94 (lines 92-95: revised version):
ï¼»20ï¼½
lines 167-168 (lines 250−251: revised version):
ï¼»16,20ï¼½
Reviewer 2 Report
I found that this paper is really interesting. The aid of a three-dimension TC reconstruction to the indocyanine green fluorescent imaging during surgery seems to be a useful innovation that can improve the chance of survival for this particular sub-group of patients suffering from HB.
However, I would like to address a few clarifications.
- In line 137 you affirm that lesions noted on TC but not palpated and not fluorescent on ICG were not removed. Were the palpated lesions removed anyway despite not ICG positive? I suppose you did remove all palpated lesions given that 23 lesions were ICG negative. Can you better clarify this notion?
- Line 145-158: would a diagram or a flowchart help the readers to better understand the process of fluorescent microscope and how many false negative and false positive lesions were found?
- Line 200-208: in the light of your findings and given that "the threshold for visualisation of intraoperative fluorescence may have not been reached", would you suggest to resect all the nodules detected by CT scan deep in the lung parenchyma in spite of the intraoperative fluorescence?
Author Response
Answer to reviewers
We greatly appreciate your suggestions.
The followings are answers to the suggestions.
- In line 137 you affirm that lesions noted on CT but not palpated and not fluorescent on ICG were not removed. Were the palpated lesions removed anyway despite not ICG positive? I suppose you did remove all palpated lesions given that 23 lesions were ICG negative. Can you better clarify this notion?
I would like to thank you for your comment. We removed all palpated lesions despite not being ICG-positive, and in fact, 23 lesions were ICG negative. I would agree with the suggestion of the reviewer and changed the term as “ All nodules palpated intraoperatively were removed despite not being ICG-positive. None of the CT-detected nodules that could not be palpated intraoperatively showed ICG fluorescence. (lines 172-175) ” And I have added Table 1.
- Line 145-158: would a diagram or a flowchart help the readers to better understand the process of fluorescent microscope and how many false negative and false positive lesions were found?
I would agree with the suggestion of the reviewer and make a table as follows (line 203).
Table 1. Sensitivity and specificity of indocyanine green (ICG) fluorescent imaging test in the pulmonary metastases resections of hepatoblastoma.
- Line 200-208: in the light of your findings and given that "the threshold for visualisation of intraoperative fluorescence may have not been reached", would you suggest to resect all the nodules detected by CT scan deep in the lung parenchyma in spite of the intraoperative fluorescence?
Thank you for your important remarks. Because ICG fluorescence has a small number of false negatives and because complete resection is important for the prognosis, we suggest that nodules noted on CT and palpated intraoperatively, with or without ICG fluorescence, should be aggressively resected even in deep lung nodules if they are not too invasive.